# LucidFusion: Generating 3D Gaussians with Arbitrary Unposed Images

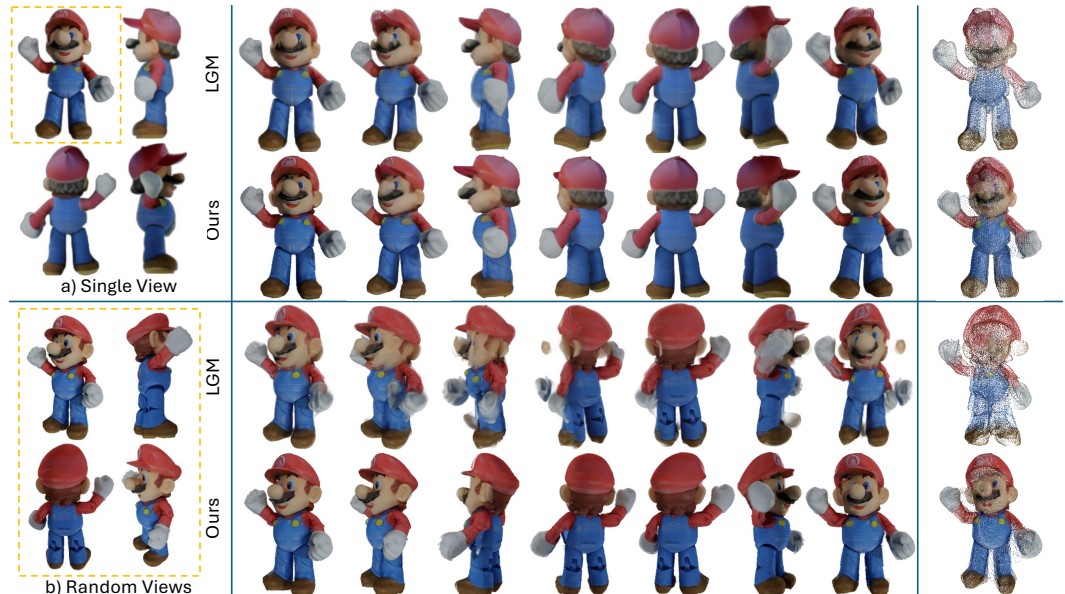

Figure 1: Our method generates high-resolution 3D Gaussians from unposed, sparse, and arbitrary numbers of multiview images. In a), we follow the current SoTA LGM (Tang et al., 2024) inference pipeline, which takes a single image as input and uses a multiview diffusion model to generate four views. In b), we compare LGM and our approach using four random input views without applying a multiview diffusion model. The yellow dashed box highlights the input views, all of which are provided without pose priors. We then compare the novel views rendered by LGM and our method in the middle column, as well as the point clouds in the last column. Note that LGM applies a Gaussian opacity threshold of 0.005 for filtering, whereas our method does not apply any post-processing thresholds.

## Abstract

Recent large reconstruction models have made notable progress in generating high-quality 3D objects from single images. However, these methods often struggle with controllability, as they lack information from multiple views, leading to incomplete or inconsistent 3D reconstructions. To address this limitation, we introduce LucidFusion, a flexible end-to-end feed-forward framework that leverages the Relative Coordinate Map (RCM). Unlike traditional methods linking images to 3D world thorough pose, LucidFusion utilizes RCM to align geometric features coherently across different views, making it highly adaptable for 3D generation from arbitrary, unposed images. Furthermore, LucidFusion seamlessly integrates with the original single-image-to-3D pipeline, producing detailed 3D Gaussians at a resolution of $512 \times 512$, making it well-suited for a wide range of applications.

## 1 Introduction

Digital 3D objects are increasingly essential in a variety of domains, enabling immersive visualization, analysis, and interaction with objects and environments that closely mimic real-world experi-

ences. These objects are foundational in fields such as architecture, animation, gaming, and virtual and augmented reality, with broad applications across industries like retail, online conferencing, and education. Despite their growing demand, producing high-quality 3D content remains a resource-intensive task, requiring substantial time, effort, and domain expertise. This challenge has catalyzed the rapid advancement of 3D content generation techniques (Mildenhall et al., 2021; Wang et al., 2021b; He et al., 2023; Hong et al., 2023; Zou et al., 2024; Huang et al., 2024; Liang et al., 2024b; Wang et al., 2024b).

Among these approaches, LRM-based approaches (Hong et al., 2023; Tochilkin et al., 2024; Zou et al., 2024) have emerged as promising solutions by training neural networks to directly regress 3D objects from single-view inputs. Recent works (Lin et al., 2023; Li et al., 2023a; Shi et al., 2023b; Tang et al., 2024; Wang et al., 2024c; Xu et al., 2024b) extend this by incorporating multi-view diffusion models to generate additional views from fixed camera positions. Although these methods achieve state-of-the-art quality with a single image input, they lack control over 3D generation due to incomplete information from unseen or occluded regions. This limitation often leads to implausible shapes and textures. A straightforward extension would be to incorporate multiple input images, which could mitigate these shortcomings by providing more comprehensive visual information for 3D object generation. However, users often face difficulties in providing images with accurately known camera poses, as estimating or calibrating such poses typically requires specialized equipment and expertise. Existing approaches are therefore limited in their ability to utilize these extra user inputs because they either require fixed camera poses as inputs (Tang et al., 2024; Wang et al., 2024c; Xu et al., 2024b) or rely on known pose information (Zhang et al., 2024). The pose information here provides essential information that links each 2D image to the 3D world space, which is important for reconstructing the unified 3D model. To mitigate this need, one could estimate camera poses first and then apply them to the input images (Wang et al., 2023b; Wu et al., 2023; Wang et al., 2024a). Unfortunately, estimating camera poses can introduce inaccuracies and increase computational overhead, which negatively impacts the efficiency and quality of 3D generation.

In response to this limitation, we propose LucidFusion, a flexible end-to-end feed-forward framework that leverages a novel representation—the *Relative Coordinate Map (RCM)*. Instead of directly linking images to world space, which typically requires pose information, our approach leverages the RCM to align geometric features consistently across different views. By transforming each view into a unified reference frame, LucidFusion enables effective 3D reconstruction without relying on explicit pose data. This bypasses the common challenges associated with pose estimation, allowing the model to handle arbitrary input images and provide better control over the 3D generation process. Additionally, RCM can easily integrate with pre-trained 2D networks, utilizing the rich priors of foundational models to enhance generalization across diverse objects and viewpoints, making the framework highly adaptable for 3D generation.

LucidFusion operates in two key stages. First, it learns to map input images to the RCM, producing a pixel-aligned point cloud representation. Second, the point cloud is refined using 3D Gaussians (Szymanowicz et al., 2024) through a rendering loss, improving fidelity and preserving object details. As demonstrated in Fig. 1 b), LucidFusion excels at generating 3D objects from arbitrary viewpoints, achieving both geometric consistency and high visual quality at 13 frames per second (FPS) for $512 \times 512$ resolution. Furthermore, when only a single view is available, LucidFusion can also leverage multi-view diffusion models for 3D generation, as shown in Fig. 1 a), making it a more flexible solution for 3D generation tasks.

In summary, our contributions are threefold:

- We train a network to map images to a novel *Relative Coordinate Map* (RCM), which embeds pixel-wise correspondences across different input views to a main view and can be converted to its point cloud representation.

- We demonstrate that RCMs can be easily obtained by fine-tuning a pre-trained 2D network to capitalize on existing 2D foundation models.

- We showcase the superior quality of our flexible method, enabling rapid 3D generation from mobile phone image captures within seconds.

## 2 RELATED WORK

### 2.1 MULTI-VIEW 3D RECONSTRUCTION

Multi-view 3D reconstruction typically relies on multi-view stereo (MVS), which reconstructs the visible surface of an object by triangulating between multiple views. MVS-based methods can be broadly classified into three categories: depth map-based methods (Campbell et al., 2008; Schönberger et al., 2016; Chang et al., 2022; Ren et al., 2023; Liang et al., 2024a), voxel grid-based methods (Kutulakos & Seitz, 2000; Yao et al., 2019; Chen et al., 2021), and point cloud-based methods (Furukawa & Ponce, 2009; Chen et al., 2019). These methods generally operate by taking multi-view images and constructing a 3D cost volume through the unprojection of 2D multi-view features into plane sweeps. However, they all depend on the availability of camera parameters with the input multi-view images, either provided during data acquisition or estimated using Structure-from-Motion (SFM) (Schonberger & Frahm, 2016; Jiang et al., 2013) for in-the-wild reconstructions. Consequently, these methods often fail when handling sparse-view inputs without known camera poses. In contrast, our approach leverages the RCM representation, enabling 3D generation from uncalibrated and unposed sparse inputs, thereby offering a robust solution for real-world applications.

### 2.2 RADIANCE FIELD RECONSTRUCTION

Neural radiance fields (NeRF) (Mildenhall et al., 2021) have recently driven significant advancements in radiance field methods, achieving state-of-the-art performance (Chen et al., 2021; Wang et al., 2021a; Ge et al., 2023). These approaches optimize radiance field representations through differentiable rendering, diverging from traditional MVS pipelines, yet they still rely on dense sampling for precise reconstruction. To address sparse-view challenges in NeRF, recent works have incorporated regularization terms (Niemeyer et al., 2022; Wang et al., 2023a) or leveraged geometric priors (Chen et al., 2021; Yang et al., 2023). However, these methods continue to require image samples with known camera poses. Another research direction explores SDS-based optimization techniques, distilling detailed information from 2D diffusion models into 3D representations (Poole et al., 2022; Wang et al., 2024b; Liang et al., 2024b), which enables the rendering of high-fidelity scenes but requires lengthy optimization for each individual scene. In contrast, our approach eliminates the need for known camera poses and operates in a feed-forward manner, supporting generalizable 3D generation without extensive optimization.

### 2.3 UNCONSTRAINED RECONSTRUCTION

Recently, the Large Reconstruction Model (LRM) (Hong et al., 2023) introduced a triplane-based approach combined with volume rendering, demonstrating that a regression model can robustly predict a neural radiance field from a single-view image, thereby relaxing the constraints on camera pose requirements. Follow-up works (Li et al., 2023a; Shi et al., 2023a;b; Xu et al., 2023; Tang et al., 2024; Zhang et al., 2024) have leveraged diffusion models to extend single-view inputs to multi-view inputs, bypassing the need for camera poses since the multi-view inputs are predicted by a pre-trained multi-view diffusion model. These feed-forward methods, trained with simple regression objectives, have achieved state-of-the-art results. However, the reliance on pretrained multi-view diffusion models, which are often overfitted to fixed camera poses (e.g., *front, back, left, right*), limits their applicability in real-world scenarios.

To address this issue, another line of research explores pose-free 3D optimization using uncalibrated images as direct input. For instance, BARF (Lin et al., 2021) employs a coarse-to-fine strategy to jointly optimize the radiance field and camera poses. NeRF– Wang et al. (2021b) enables 3D scene reconstruction and novel view synthesis without requiring known camera poses. Other approaches (Bian et al., 2023; Meuleman et al., 2023; Fu et al., 2023) utilize depth information to constrain the optimization process. More recent work, PF-LRM Wang et al. (2023b), predicts poses from multi-view images for 3D reconstruction. However, these methods typically require test-time optimization or are limited to a small number of input views. Another line of work working on an intermediate 3D representation to bridge 2D and 3D, SweetDreamer (Li et al., 2023b) and CRM (Wang et al., 2024c) leverages pointmap for geometry regularization. Dust3R (Wang et al., 2024a) and InstantSplat (Fan et al., 2024) employ pointmap representations for pose estimation from image pairs,

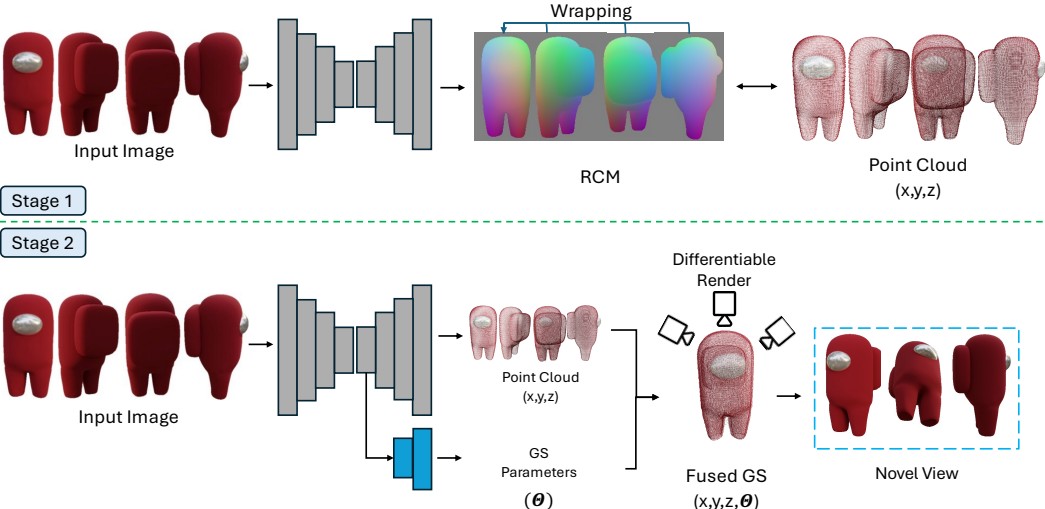

Figure 2: Pipeline Overview of LucidFusion. Our framework processes a set of sparse, unposed multi-view images as input. These images are concatenated along the width dimension and passed through the Stable Diffusion model in a feedforward manner. The model predicts the RCM representation for the input images. Additionally, the feature map from the final layer of the VAE is fed into a decoder network to predict Gaussian parameters. The RCM representation and the predicted Gaussian parameters are then fused and passed to the Gaussian renderer to generate novel views for supervision.

but they are restricted to pairs of images and necessitate test-time optimization for global alignment across all input views. Concurrent work (Xu et al., 2024a) also leverages a coordinate map representation with a generative diffusion model but relies on an additional PnP solver for refinement and is limited to no more than 6 views. In contrast, our approach utilizes an intermediate pointmap representation within a regression framework that can be directly fed into the Gaussian renderer without the need for additional refinement steps. Moreover, our regression-based method accommodates an arbitrary number of unposed inputs for 3D reconstruction, significantly enhancing rendering efficiency and producing high-quality results suitable for practical applications.

## 3 METHOD

Lucidfusion is a feed-forward 3D generation model, which takes one to $N$ input images and effectively infers their 3D Gaussians, as shown in Fig. 2. LucidFusion can reconstruct the 3D object effectively from input images, regardless of whether they are captured around the object or generated by a multi-view diffusion model. In this section, we first explain our motivation for using the RCM representation in Sec. 3.1, then define our proposed relative coordinate map (RCM) in Sec. 3.2. We introduce using 3D Gaussian refinement in Sec. 3.3, and finally discuss the loss function in Sec. 3.4.

### 3.1 PRELIMINARY

Lifting the condition from a single image to multiple images introduces several challenges. We abstract the 3D generation problem as a mapping task: with a single image, the focus is on extracting geometric information for generation, but with multiple images, both mapping and scaling issues arise. This mapping can be explicit, as in traditional MVS-based methods (Yao et al., 2018; Chen et al., 2021), or implicit, as in LRM-based approaches (Hong et al., 2023). However, both approaches require pose estimation, where posed images must either be estimated or fixed at specific viewpoints, limiting the pipeline's flexibility. To address this, we propose a novel method that performs the mapping end-to-end without relying on explicit pose information.

## 3.2 RELATIVE COORDINATE MAP

We argue that explicitly providing pose information is not necessary in this task. The key is to ensure consistent geometric feature estimation across different viewpoints and maintain scale-wrapping relationships. Based on this idea, we propose fusing the geometric features with multi-view image inputs. Specifically, we wrap the object's location by transforming each individual view's coordinates to align with a selected main view's coordinate system. We refer to this new presentation as the Relative Coordinate Map (RCM), which we formally define in the following section.

Given a set of $N$ input images, denoted as $I_i \in \mathbb{R}^{H \times W \times 3}$, and the RCM is defined as $M_i \in \mathbb{R}^{H \times W \times 3}$, where $i = 1, 2, 3, \ldots, N$, and $H$ and $W$ represent the height and width of the corresponding image. The RCM is therefore the corresponding coordinates of each image pixel in the 3D space. To facilitate the model's learning of these coordinates from arbitrary viewing directions, we reproject all $N$ input images into the coordinate system of one of the input views, selected randomly. This process allows the model to generalize to varying viewpoints. Given the camera pose $P_i \in \mathbb{R}^{4 \times 4}$ and intrinsic $K \in \mathbb{R}^{4 \times 4}$ in homogeneous form, along with the depth map $D_i \in \mathbb{R}^{H \times W}$, we randomly choose one of the $P_i$ as main camera pose $P_{main}$. The main camera's RCM is thus defined as:

$$M_{main} = P_{main} P_{main}^{-1} K^{-1} * D_{main}. \tag{1}$$

Therefore, the RCM for the main camera is formulated as:

$$M_{main} = K^{-1} * D_{main}, \tag{2}$$

as defined in its own camera coordinate frame. The remaining input views are reprojected into the main camera's coordinates as

$$M_j = P_{main} P_j^{-1} K^{-1} * D_j, \tag{3}$$

where $j = 1, 2, 3, \ldots, N - 1$. The RCM values are constrained within the range of $[-1, 1]$. To enhance 3D consistency across multi-view inputs, we concatenate the input images along the width dimension $W$, enabling the model to leverage self-attention mechanisms to explore multiple views simultaneously.

The RCM representation offers several key advantages. First, as an image-based representation, it benefits from pre-trained foundation models, thereby simplifying the learning process. Second, RCM maintains a one-to-one mapping between image pixels and the 3D seen surface, effectively capturing the geometric information of objects as a point cloud. Finally, by concatenating multiple input images into a unified input, the model facilitates geometric scale interactions and alignment across viewpoints through a self-attention mechanism, ensuring 3D consistency in the coordinate map across different viewpoints.

Specifically, to obtain the RCM representation, we train a network $E$ that processes $N$ RGB images $I_i \in \mathbb{R}^{H \times W \times 3}$, predicting the corresponding RCMs $M_i \in \mathbb{R}^{H \times W \times 3}$, where $i = 1, 2, 3, \ldots, N$. Additionally, We extract the intermediate feature map $f_i \in \mathbb{R}^{\frac{H}{8} \times \frac{W}{8} \times l}$ from network $E$ and pass it to the decoder network $G$ to predict 3D Gaussians (Szymanowicz et al., 2024) for rendering, which we will discuss in details in Sec. 3.3. Formally, this process can be defined as:

$$M_i, f_i = E(I_i). \tag{4}$$

The network $E$ maps the RGB inputs to their corresponding RCM representations, enabling the use of a generic 2D model for the network $E$ without the need for 3D priors. The RCM $M_i$ represents the 3D surface visible in the input images, maintaining per-pixel alignment with the corresponding RGBs.

## 3.3 3D GAUSSIAN REFINEMENT

We observed that the point cloud obtained from the RCM representation in Sec. 3.2 is noisy, as shown in Fig. 3.We attribute this effect to two primary factors. First, the RCM is regressed solely from a set of 2D images without any explicit 3D prior information, making it challenging to maintain 3D consistency. Second, due to the inherent limitation of convolutional models, the RCM struggles to accurately capture object boundaries, often resulting in partial misalignment. To refine this noisy point cloud, we adopted 3D Gaussians (Szymanowicz et al., 2024), which complement the RCM representation by introducing global 3D awareness and improving overall geometric consistency.

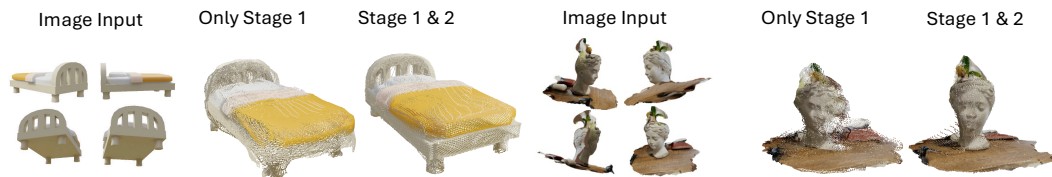

Figure 3: Point Cloud comparison, a) stage 1 only, b) stage 1 with stage 2 refinement.

Specifically, we use the noisy point cloud as initialization and refine it through rendering loss, with the refined point cloud shown in Fig. 3. We utilize the feature map $f_i$ obtained from Eq. 4 and employ the decoder $G$ to transform it into 3D Gaussian parameters. This process is formally defined as:

$$\Theta_i = G(f_i), \tag{5}$$

where the $\Theta_i$ denotes the 11-channel 3D Gaussian parameters: 3-channel RGB variation $\delta^c$, 3-channel scale $s$, 4-channel rotation quaternion $rot$, and 1-channel opacity $\sigma$ for each input image $I_i$. Given that the RCM $M_i$ represents the corresponding coordinates of the object, the final input to the 3D Gaussian render is defined as:

$$\Theta_i = (M_i, I_i + \delta_i^c, s_i, rot_i, \sigma_i), \tag{6}$$

where $i = 1, 2, 3, \ldots, N$. Consequently, we generate a total of $\{N \times H \times W\}$ 3D Gaussians $\Theta$. It is important to note that the number of output Gaussians $\Theta$ scales proportionally with the number of input views.

**Disccusion.** Recent methods (Li et al., 2023b; Wang et al., 2024c) have explored canonical coordinate maps (CCM) as a representation for 3D objects. However, these methods focus solely on geometric representation without addressing the unification of multiple viewpoints. When regressing CCM from multi-view inputs, the model operates under the world coordinate convention and must simultaneously infer object's orientation and geometry. This is reflected in visualization, where the same body parts should retain consistent colors across all views. For instance, as shown in the middle row of Fig. 4, the sheep's head and tail should appear the same color across all views. This semantic information is crucial for indicating an object's orientation

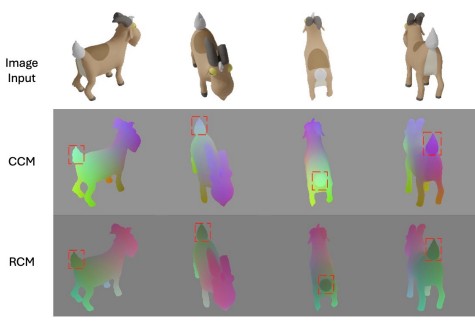

Figure 4: We compare CCM and RCM to evaluate their performance from 2D images.

in world space. Misalignment in color suggests that the model has failed to correctly map the object from the 2D multi-view inputs.

However, this task is extremely challenging given only a few 2D images as input, because the model must: a) maintain 3D consistency across input views, and b) learn the object's orientation, recognizing that it is the same object across all views rather than a different object with varying orientations. In contrast, our proposed RCM representation resolves these orientation ambiguities by transforming the coordinates into a unified coordinate system. As shown in the bottom row of Fig. 4, our approach explicitly addresses the issue of orientation, making it more suitable for our needs.

### 3.4 LOSS FUNCTIONS

In stage 1, we take $N$ RGB input views and predict their corresponding RCMs, supervising them with ground truth RCMs obtained using Eq. 3. We minimize the loss between predicted RCMs $\hat{M}_i$ and ground truth RCMs $M_i$ using the Mean Square Error (MSE) loss:

$$L_{rcm} = \frac{1}{N} \sum_{i=1}^{N} L_{MSE}(\hat{M}_i, M_i). \tag{7}$$

In stage 2, we utilize the predicted Gaussian splats from Eq. 6 and employ the differentiable renderer from Kerbl et al. (2023) to render $V$ supervision views. For this stage, we adopt a combination of

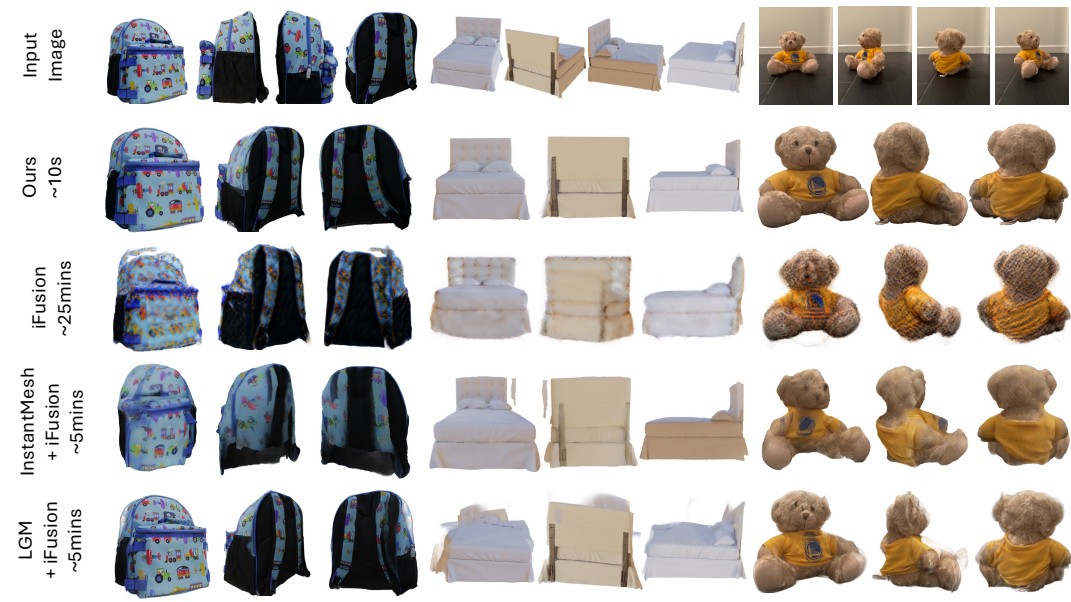

Figure 5: Qualitative comparison with baseline models: iFusion (Wu et al., 2023), InstantMesh (Xu et al., 2024b) and LGM (Tang et al., 2024).

MSE loss, SSIM loss from Kerbl et al. (2023), and VGG-based LPIPS loss (Zhang et al., 2018) to the RGB image:

$$L_{rgb} = (1 - \lambda)L_{MSE}(\hat{I}_i, I_i) + \lambda L_{SSIM}(\hat{I}_i, I_i) + L_{LIPIS}(\hat{I}_i, I_i), \tag{8}$$

where $\lambda$ is set to 0.2 as Kerbl et al. (2023). Additionally, to expedite convergence, we apply MSE loss to the alpha channel image, as proposed in Tang et al. (2024):

$$L_{\alpha} = L_{MSE}(\hat{I}_i^{\alpha}, I_i^{\alpha}). \tag{9}$$

Thus, the final loss for stage 2 is:

$$L = L_{rgb} + L_{\alpha}. \tag{10}$$

## 4 EXPERIMENT RESULTS

### 4.1 IMPLEMENTATION DETAILS

For stage 1, similar to concurrent work (He et al., 2024), we empirically found that using a pre-trained Stable Diffusion model (Rombach et al., 2022) in a purely feedforward manner, bypassing the need for multiple diffusion steps achieves the best result. For Stage 2, the SD VAE decoder is adapted to generate Gaussian splats. We conducted the training on 8 NVIDIA A100 (80G) GPUs for both Stage 1 and Stage 2. For more details please see the appendix.

| | GSO | | | ABO | | |
|---|---|---|---|---|---|---|
| | PSNR↑ | SSIM↑ | LPIPS↓ | PSNR↑ | SSIM↑ | LPIPS↓ |
| iFusion | 17.21 | 0.852 | 0.180 | 17.54 | 0.853 | 0.180 |
| LGM+iFusion | 19.61 | 0.872 | 0.131 | 19.89 | 0.873 | 0.131 |
| InstantMesh+iFusion | 20.75 | 0.894 | 0.127 | 20.98 | 0.901 | 0.129 |
| Ours | **25.97** | **0.930** | **0.070** | **25.98** | **0.917** | **0.088** |

Table 1: Performance comparison against baselines on GSO and ABO for 4 views input.

| | GSO | | | ABO | | |
|---|---|---|---|---|---|---|
| | PSNR↑ | SSIM↑ | LPIPS↓ | PSNR↑ | SSIM↑ | LPIPS↓ |
| CRM | 16.74 | 0.858 | 0.177 | 19.23 | 0.871 | 0.169 |
| LGM | 14.31 | 0.824 | 0.186 | 16.03 | 0.861 | 0.181 |
| InstantMesh | 16.84 | **0.864** | 0.177 | **19.73** | 0.873 | 0.168 |
| Ours | **16.91** | 0.862 | **0.177** | 19.51 | **0.873** | **0.168** |

Table 2: Performance comparison against baselines on GSO and ABO for single-image-to-3D setting.

### 4.2 QUALITATIVE COMPARISON

We first compare LucidFusion aginst baseline models under a sparse input view setting, where the sparse input views without pose are from GSO (Downs et al., 2022), ABO (Collins et al., 2022)

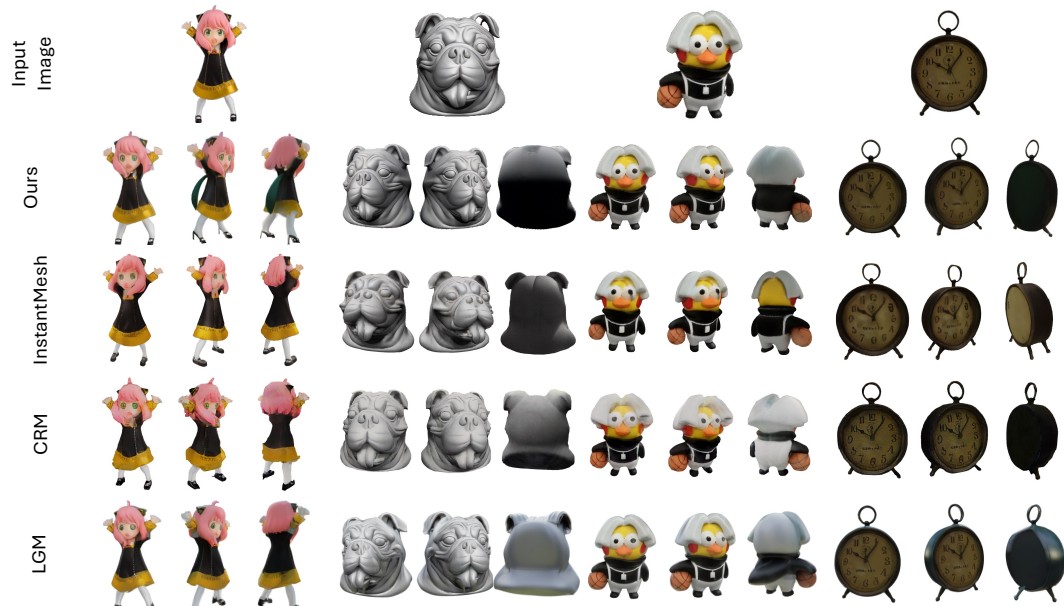

Figure 6: Qualitative comparison with baseline model InstantMesh (Xu et al., 2024b), CRM (Wang et al., 2024c) and LGM (Tang et al., 2024) under standard single-image-to-3D paradim.

and an iPhone capture. Since the input poses are unknown, we first compare ours with a recent open-source pose-free reconstruction approach iFusion (Wu et al., 2023) as our baseline. Moreover, we use the estimated pose from iFusion to the current SoTA reconstruction methods, LGM and InstantMesh to achieve pose-free sparse view generation for comparison. All baseline models are used from their official implementations. As shown in Fig. 5, it can be seen that our proposed Lucidfusion outperforms other baselines with better geometry and visual quality. Also, our method does not assume pose during inference time, thus is much faster than other baselines that depend on pose estimation.

We also follow the standard single-image-to-3D paradigm to evaluate our method, demonstrate the generalizability of our method that can also combine with multi-view diffusion models. Specifically, we use CRM's pixel diffusion to generate multi-views with a different seed. As shown in Fig. 6. Our method can utilize the multi-view diffusion model and faithfully produce high-resolution 3D Gaussians.

In Fig. 11, we demonstrate our model's generalization ability across different data sources. Our model produces high-quality 3D Gaussian at a resolution of $512 \times 512$. We showcase two real-world, in-the-wild captures using a handheld iPhone 15, where our method successfully reconstructs the objects while preserving content from the unposed sparse multi-view inputs.

### 4.3 QUANTITATIVE COMPARISON

We evaluate our method against baselines by conducting two different experiments on the GSO and ABO datasets. For each dataset, we randomly sample 100 objects and render 24 images at elevations of $5°$ and $20°$. We use images from the $20°$ elevation set as input and evaluate the model's performance on the $5°$ elevation set. We consider two different input selection strategies: a) four random input views without pose, and b) single-image-to-3D. To ensure a fair comparison, we use iFusion (Wu et al., 2023) to estimate pose in a) for baselines that require pose. For the experiment in b), we utilize the multi-view diffusion model from CRM (Wang et al., 2024c) with a different seed. As shown in Tab. 1, our model consistently outperforms the baselines across all metrics by a large margin. Notably, iFusion (Wu et al., 2023) is an optimization-based method, introducing a 5-minute overhead for pose estimation, whereas our method does not require pose and achieves 13 FPS. As shown in Tab. 2, our approach works effectively within the existing single-image-to-3D paradigm, delivering on-par performance with current baselines.

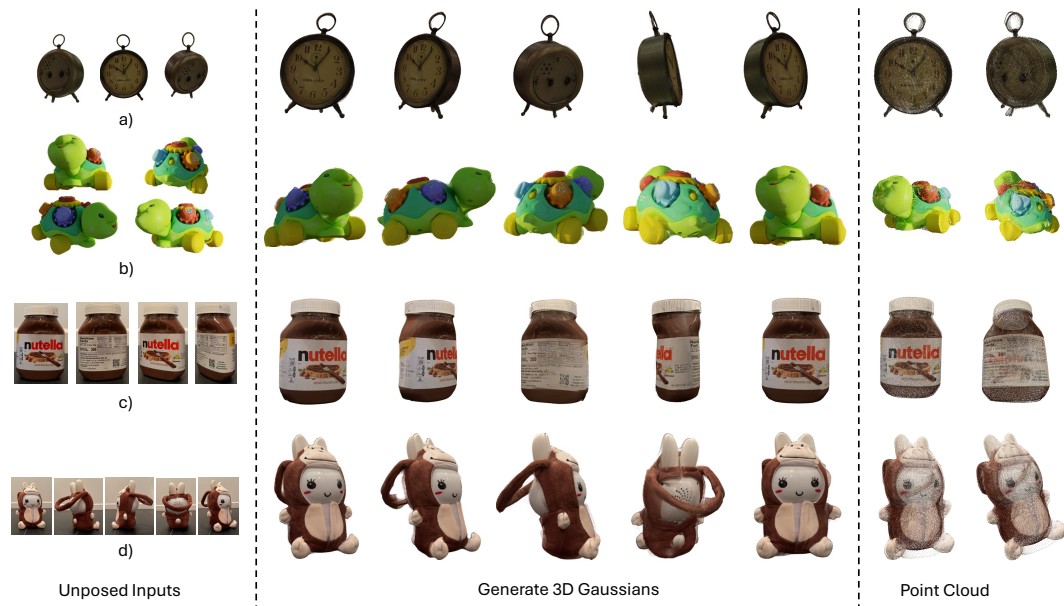

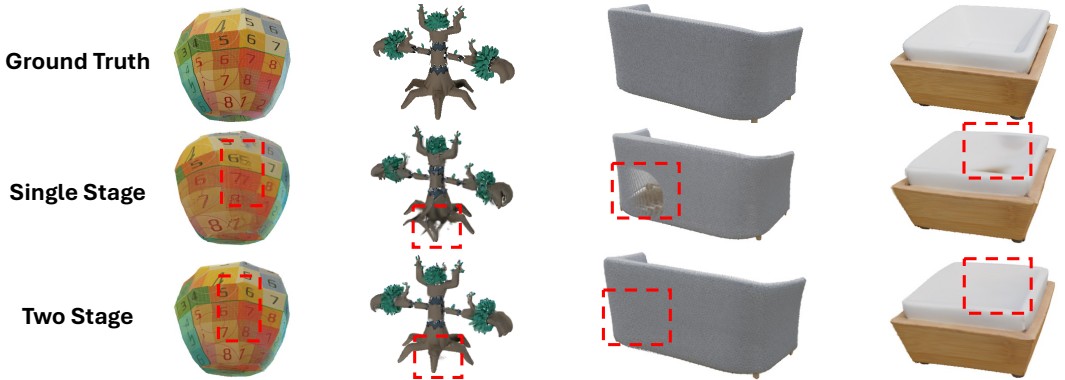

Figure 7: Cross-dataset generalization to unseen objects, the first column shows the input unposed sparse views, the second column shows the generated Gaussian novel views at resolution $512 \times 512$, and the final column shows the RCM representation project to its point cloud. a) and b) Multi-view image from GSO datasets (Downs et al., 2022). c) and d) IPhone captured objects, the multi-view image was captured using a handheld iPhone 15pro, we first removed the background and recentered the object before passing it into our pipeline.

Figure 8: Comparison with single and two-stage training. Without pertaining to RCM representation, the model struggles to correctly predict Gaussian locations, resulting in mismatches or empty holes in rendering.

## 4.4 ABLATION STUDY

**Single Stage Training.** We train the model without Stage 1 RCM training, as depicted in Fig. 2, meaning the entire model is supervised solely by rendering loss. The visualization results in Fig. 8 demonstrate that without Stage 1 training, the model struggles to accurately predict object coordinates, leading to ambiguous rendering outcomes. The two-stage model consistently outperforms the single-stage model across all evaluation metrics. Please find more results in the appendix.

**Number of Views.** We evaluate our model with varying numbers of input views, ranging from $i = 1, 2, \ldots, 8$ and report the performance in Tab. 3. As described in Sec. 3, our model is capable of handling a varying number of input views for 3D reconstruction.

**Training with Random Views.** We evaluate our model's performance in Stage 2 training under fixed and random input view settings. In the fixed setting, we train for 20 epochs with 5 input

Figure 9: Visualization of LucidFusion for single image input results without multi-view diffusion model.

Figure 10: Visualization of different training strategy results with single image input.

| # of test view | Fixed Strategy | | | Random Strategy | | |
|---|---|---|---|---|---|---|
| | PSNR↑ | SSIM↑ | LPIPS↓ | PSNR↑ | SSIM↑ | LPIPS↓ |
| 1 | 17.36 | 0.860 | 0.160 | 21.83 | 0.904 | 0.112 |
| 2 | 21.83 | 0.904 | 0.080 | 23.62 | 0.915 | 0.091 |
| 4 | 25.95 | 0.932 | 0.070 | 25.97 | 0.930 | 0.070 |
| 6 | 26.15 | 0.933 | 0.070 | 26.11 | 0.931 | 0.070 |
| 8 | 26.22 | 0.933 | 0.070 | **26.25** | **0.933** | **0.069** |

Table 3: Comparison between training with a random number of input views. The result shows that different strategies perform similarly when we have sufficient input views. However, a random number of training views strategy outperforms a fixed one by a large margin when input views are limited.

views, while in the random setting, we randomly sample between 1 and 5 views per batch over 20 epochs. The results, shown in Tab. 3, indicate that the model trained with random input views performs better when the number of views is limited. Notably, with only a single input view, the model trained with random views generates less blurriness and fewer empty textures in unseen regions. However, when the input views provide sufficient coverage of the object, both models exhibit comparable performance. As demonstrated in Fig. 10, the model trained with a fixed number of views struggles to predict unseen regions, whereas the random view training strategy still produces reasonable predictions for those regions. It is important to note that single-image reconstruction is inherently ill-posed; while the model can faithfully reconstruct seen regions, it may fail in unseen areas. Nonetheless, LucidFusion provides reliable predictions under such conditions, showcasing its superior performance. Additional results for single-image input across various data sources are presented in Fig. 9.

## 4.5 LIMITATION

Despite the promising results, our model has some limitations. First, it can only render objects positioned at the center of the scene, without backgrounds. We hypothesize that incorporating background information into the RCM representation during training could address this issue, which we leave for future work. Additionally, our current model is trained on Objaverse data with a fixed field of view (FoV) of $30°$. As a result, objects that deviate significantly from this setting may exhibit shape distortions. Future work could explore training on a wider variety of settings and FoVs to enhance the robustness of the RCM representation.

## 5 CONCLUSION

In this work, we propose LucidFusion, a flexible end-to-end feed-forward framework that leverages the Relative Coordinate Map (RCM), a novel representation designed to align geometric features coherently across different views. Our model employs a Stable Diffusion to map RGB inputs to RCM representations in a feedforward manner and uses an efficient Gaussian renderer to produce high-resolution 3D content. This approach ensures robust control over the 3D generation process, delivering high-quality outputs across a range of scenarios. LucidFusion also integrates seamlessly with the original single-image-to-3D pipeline, making it a versatile tool for 3D object generation. We believe this work will open new avenues for future research in the field of 3D generation.

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

# A APPENDIX

## A.1 DATASETS

The Objaverse dataset (Deitke et al., 2023) contains approximately 800,000 shapes. Given the presence of numerous low-quality 3D models in the dataset (e.g., single planes, partial scans), we employed crowd workers to manually filter the dataset, focusing on objects rather than other assets like scans of large scenes or buildings. Additionally, we excluded objects that rendered predominantly in white, as this typically indicates missing textures. After filtering, our final dataset comprises approximately 98K 3D objects.

Using Blender, we generated synthetic images from the 3D meshes and extracted various useful annotations, including depth maps, camera intrinsics, poses, and images. We rendered these objects using a field of view (FOV) of $30°$ and elevations of $-20°$, $5°$ and $20°$, along with front views. Each elevation setting was rendered with 24 images, while the front view was rendered with 18 images, resulting in a total of 90 images per object. During training, $N$ views are randomly sampled from these 90 images. The rendered images have a resolution of $512 \times 512$ and are generated under uniform lighting conditions.

We evaluate our model on two 3D object datasets, Google Scanned Object (GSO) (Downs et al., 2022) and Amazon Berkeley Objects (ABO) (Collins et al., 2022). For each dataset, we randomly sample 100 objects and render 24 images at elevations of $5°$ and $20°$. We use images from the $20°$ elevation set as input and evaluate the model's performance on the $5°$ elevation set.

| | w/o stage 1 (SD) | w. stage 1 (SD) | w. stage 1 (DPT) | w. stage 1 (SVD) |
|---|---|---|---|---|
| PSNR↑ | 24.15 | **25.97** | 24.15 | 23.96 |
| SSIM↑ | 0.916 | **0.930** | 0.917 | 0.916 |
| LPIPS↓ | 0.080 | **0.070** | 0.091 | 0.088 |

Table 4: Performance comparison for Stage 1 with different encoders, tested on GSO dataset with sparse 4 view setting.

## A.2 MORE IMPLEMENTATION DETAILS

**Stage 1.** Similar to concurrent work (He et al., 2024), we empirically found that using a pretrained Stable Diffusion model (Rombach et al., 2022) in a purely feedforward manner, bypassing the need for multiple diffusion steps achieves the best result, as shown in Tab. 4. The feature map $f$ is extracted before the final output layer and used by the decoder to generate Gaussian splats in Stage 2. The feature map $f$ has a shape of $\{N, 320, \frac{H}{8}, \frac{W}{8}\}$, where $H$ and $W$ denote the image resolution. During training, we unfreeze the VAE decoder and UNet components, training the SD model in a feedforward manner without utilizing diffusion steps. Specifically, we set the text prompt to an empty string ("") and use $t = 999$ for the scheduler. The RGB input views are set to 5.

**Stage 2.** For Stage 2, the SD VAE decoder is adapted to generate Gaussian splats. We modify the SD VAE decoder to accept a channel size of 320 and output 11-channel Gaussian splat predictions, which are then processed by a Gaussian renderer to generate supervision views. During training, we randomly sample between 1 and 5 input views and render additional novel views to produce a total of 8 views for supervision. The SD and VAE decoder are trained simultaneously using only the rendering loss.

We conducted the training on 8 NVIDIA A100 (80G) GPUs for both Stage 1 and Stage 2. In Stage 1, we train the model on images with a resolution of $256 \times 256$. The batch size for Stage 1 is set to 4 per GPU, resulting in an effective batch size of 32. We train for 40 epochs and Stage 1 training takes approximately 3 days. For Stage 2, we use a batch size of 2 per GPU, resulting in an effective batch size of 16, with training taking around 4 days for 20 epochs. The output 3D Gaussians are rendered

at a resolution of $512 \times 512$. We utilize the AdamW optimizer (Loshchilov & Hutter, 2017) with a learning rate of $3 \times 10^{-5}$ for stage 1 and 2.

## A.3 MORE VISUALIZATION RESULTS

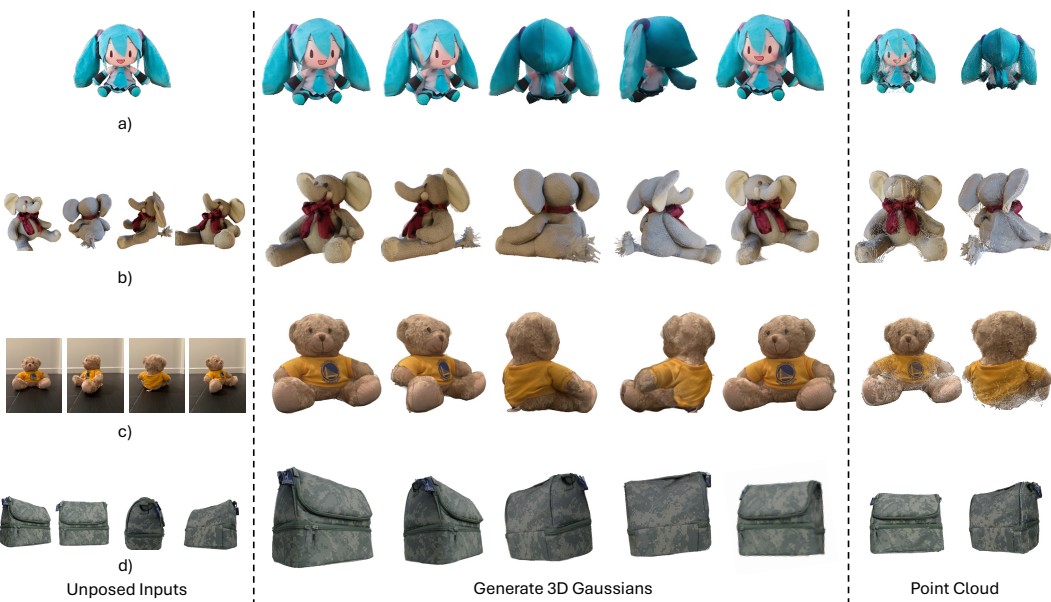

Figure 11: More visualization results for cross-dataset generalization, a) We demonstrate that our model can operate under single-view-to-3D with multi-view diffusion model. Moreover, b),c) and d) shows that our model generalizes effectively to varying numbers of unposed input views across different data sources.

## A.4 VISUALIZATION OF RCM

We visualize the predicted RCM map from input images, as shown in Fig. 12. Starting with a set of 2D images, we predict their corresponding RCM representation within the range of $[-1, 1]$. Since the RCM representation is per-pixel aligned with the input images, we concatenate them into a shape of $[N, 6, 3]$, where $N$ is the total number of points, defined as $H \times W \times V$, with $H, W$, and $V$ representing the image height, width, and number of input views, respectively.

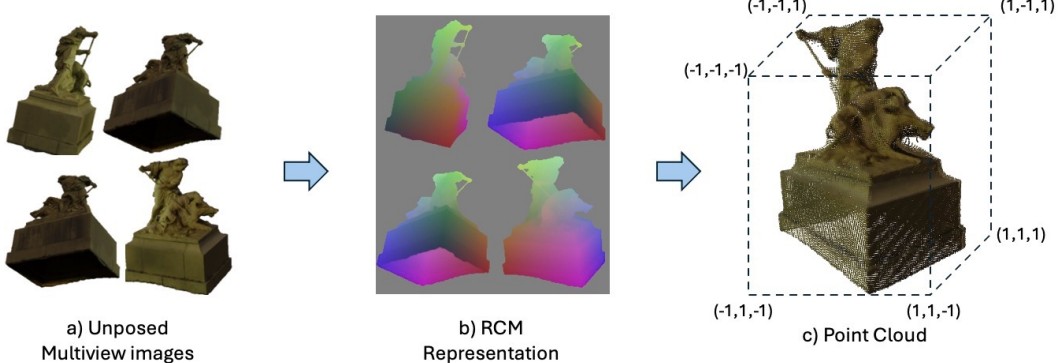

Figure 12: **Relative Coordinate Map.** Visualization of Relative Coordinate Map (RCM).

## A.5 DIFFERENT STAGE 1 NETWORK

We evaluate the performance of different stage 1 network on 100 randomly selected objects from GSO (Downs et al., 2022) by training each of them 20 epochs and report their performance in Tab. 4. The Stable Diffusion (SD) (Rombach et al., 2022) consistently outperforms other stage 1 networks.

