# OpenReview forum: "LucidFusion: Generating 3D Gaussians with Arbitrary Unposed Images"
_ICLR.cc/2025/Conference — ICLR 2025 Conference Withdrawn Submission_

### Official Review · Reviewer_WgV6 · 2024-11-02

**Soundness:** 2
**Presentation:** 2
**Contribution:** 2
**Rating:** 3
**Confidence:** 5

**Summary:**

The paper introduces LucidFusion, a feed-forward framework for generating high-resolution 3D Gaussians from arbitrary, unposed multi-view images. LucidFusion leverages a Relative Coordinate Map (RCM) to align geometric features across different views without relying on explicit pose information. The framework consists of two stages: mapping input images to an RCM and refining the point cloud representation using 3D Gaussians for improved fidelity. The key contributions are: training a network to map images to the RCM, showcasing the ease of obtaining RCMs by fine-tuning pre-trained 2D diffusion models, and demonstrating the method's 3D generation capability from multi-view renderings.

**Strengths:**

- The proposed method eliminates the need for input poses, demonstrating significant advantages over previous pose-dependent LRMs.
- Leveraging 2D diffusion prior for RCM prediction is interesting, benefiting the generalization capability of the framework.
- Thorough ablation studies were conducted to prove the effectiveness of each model component.

**Weaknesses:**

- Incomplete experiments.
The experimental results are not strong enough to support the paper's calims.
(1) The quality of RCM prediction. The paper only reports view synthesis metrics, i.e., PSNR/SSIM/LPIPS, lacking evaluations on the quality of the 3D point maps predicted by the diffution model. I think some 3D metrics such as Chamfer Distance with the ground truth point cloud are required.
(2) Incomplete and potentially unfair comparison with baseline methods. The results of Instantmesh and LGM in Figure 5 and Table 1 are based on poses predicted by iFusion, which maybe bad-quality. I think the authors should include the results of feeding ground truth poses into them as a reference. In fact, the 3D point maps predicted by LucidFusion enables camera pose estimation using a PnP solver, similar to the practice of SpaRP [1]. I think the authors should try feeding the pose predictions of LucidFusion to the baseline methods and include the results in the experiments.
(3) Robustness to random seeds. The proposed method leverages a diffusion model to predict RCMs, making the output dependent on random seeds. Is the model able to produce faithful RCMs consistently with different seeds? I think the author should discuss this issue in the paper.
(4) Limited test set size. The experiments are conducted on only 100 objects from GSO and ABO respectively, which is too small to make the reported metrics convincing.

- Incremental technical contribution and unsatisfying performance.
Predicting 3D coordinate maps from images using a pretrained 2D diffusion model for sparse-view reconstruction is not novel and has been explored by prior works, such as CRM and SpaRP. This paper simply introduces a Gausian prediction head to the SD VAE decoder, showing incremental technical contribution. Besides, the video visualizations in the supplemental material demonstrate significant shape distortions (e.g., the "bell" sample) and multi-view inconsistency (e.g., the "bear" and "mario" samples). The geometry and texture of these samples are not challenging, but the quality of the results are limited, thus I hold a negative perspective on the model's practicality and performance on more challenging objects.

- Minor.
Which diffusion model does the "DPT" in Table 4 refer to? I cannot find a reference in the paper.

[1] Xu, Chao, et al. "Sparp: Fast 3d object reconstruction and pose estimation from sparse views." arXiv preprint arXiv:2408.10195 (2024).

**Questions:**

Please refer to the "Weakness" part for my concerns. The overall technical contribution and performance of this work cannot reach my acceptance threshold for now, but I'm still open to adjust my final rating according to the author's reponses to the aforementioned issues.

---

### Official Review · Reviewer_rWJH · 2024-11-03

**Soundness:** 2
**Presentation:** 2
**Contribution:** 2
**Rating:** 3
**Confidence:** 4

**Summary:**

This paper addresses the challenge of generating high-quality 3D objects from single or sparse multi-view images without relying on camera pose information. The authors propose the LucidFusion framework, which leverages a novel Relative Coordinate Map (RCM) to achieve cross-view geometric alignment without explicit pose data and incorporates a 3D Gaussian refinement step to enhance detail fidelity. Experimental results demonstrate that LucidFusion can produce consistent, high-quality 3D reconstructions solely from limited or unposed images, significantly improving flexibility and adaptability for real-world applications.

**Strengths:**

1. The authors present an RCM approach that operates without requiring view pose information, achieving robust cross-view geometric consistency.

2. The paper is well-written, with a clear structure and logical progression.

3. The experimental results demonstrate the effectiveness of the proposed approach.

**Weaknesses:**

1. The method’s contributions are largely incremental, with RCM-based alignment and VAE decoding closely paralleling existing approaches (e.g., Dust3r and LGM), offering only minor improvements in consistency.

2. Experimental details are lacking, with limited comparisons, few ablation studies, and minimal analysis, which weakens the clarity and persuasiveness of the results.

**Questions:**

1. How significant is the improvement in point cloud consistency achieved by RCM compared to existing methods like Dust3r? Is there a quantitative measure for this improvement, and is there a detailed analysis explaining RCM’s mixed performance, especially in Stage 1?

2. What is the rationale behind using a VAE decoder to predict Gaussian parameters in the final layer? Does this design offer measurable benefits over LGM’s approach, or is it primarily a way to reinforce consistency through feature sharing? Could alternative designs yield better 3D quality?

3. Where is the depth  D  in the training data derived from? Is it obtained from external datasets, estimated through another model, or synthesized? Additionally, how reliable is this depth information, and how does its accuracy impact the training and final 3D reconstruction quality?

4. Are the selected comparative methods sufficiently representative, especially given the method’s pose-free nature? Would additional comparisons with other pose-free 3D generation approaches strengthen the experimental conclusions?

---

### Official Review · Reviewer_yRQ4 · 2024-11-04

**Soundness:** 2
**Presentation:** 2
**Contribution:** 2
**Rating:** 3
**Confidence:** 4

**Summary:**

This paper focuses on 3D reconstruction from unposed sparse-view images. It proposes finetuning a pretrained 2D diffusion model (e.g., stable diffusion) to output point coordinate maps and other Gaussian parameters for the input views, which can then be used for 3D reconstruction. The point coordinate maps are defined in the camera frame of the main view. The authors also propose refining the Gaussians obtained from the feedforward model through a refinement step using rendering losses.

**Strengths:**

1. The idea of fine-tuning stable diffusion to output point maps is interesting for solving the unposed sparse-view reconstruction problem.

2. The authors demonstrate that the proposed method outperforms the baseline method, iFusion, and provide some ablation studies.

**Weaknesses:**

1. Training a network to output point maps relative to the main (or first) view is not a brand new idea in unposed sparse-view reconstruction [1, 2]. Although the authors claim [2] to be a concurrent work, it was actually submitted to ECCV 2024, which weakens this assertion.

2. While the paper claims to handle unposed sparse-view reconstruction, the implementation and evaluation details suggest that the proposed method does not truly accommodate arbitrary random poses. For instance, the training data is limited to a fixed set of elevation angles {-20°, 0°, 20°}. Similarly, during evaluation, the images are rendered from these same angles. This assumption is problematic and could be seen as misleading, as it deviates from the arbitrary, random camera poses expected in this task.

3. The method also assumes a fixed field of view (FoV), a fixed up direction, and a look at direction to the object’s center, which may differ significantly from real-world captured images. It is strongly recommended that the authors evaluate their method on real-world datasets like OmniObject3D and MVImageNet, which cover a broader range of camera poses.

4. The method only reports neural view synthesis (NVS) metrics (e.g., LPIPS), without including any 3D metrics. This may be unfair to feed-forward methods (e.g., LGM and InstantMesh) as the proposed method incorporates additional refinement with rendering losses. It would be fairer to include lightweight optimization for the baseline methods as well.

5. The proposed method relies on input views covering all regions of the object. If the sparse-view input images fail to capture all regions, the method may struggle to generate unseen areas.

6. The presentation of the paper could be greatly improved, as many details are currently unclear or confusing. Some notable examples include:

   (a) In Figure 2, "Stage 2" appears to refer to "training a network to predict other Gaussian parameters" in addition to RCM. However, in Figure 3, Stage 2 seems to refer to optimization with rendering losses as a refinement step.

   (b) It is unclear how the authors use Stable Diffusion to predict RCM and other Gaussian parameters. Stable Diffusion consists of multiple submodules (e.g., VAE encoder, UNet, VAE decoder). Which submodules are used for predicting RCM? Among them, which are frozen, and which are fine-tuned? Similarly, which submodules are used for predicting Gaussian parameters? Why do these require different training recipes and two-stage training? Can the RCM and other Gaussian parameters be unified in a single, unified training process?


[1] Wang, Shuzhe, Vincent Leroy, Yohann Cabon, Boris Chidlovskii, and Jerome Revaud. "Dust3r: Geometric 3D Vision Made Easy." In *Proceedings of the IEEE/CVF Conference on Computer Vision and Pattern Recognition*, pp. 20697–20709. 2024.

[2] Xu, Chao, Ang Li, Linghao Chen, Yulin Liu, Ruoxi Shi, Hao Su, and Minghua Liu. "Sparp: Fast 3D Object Reconstruction and Pose Estimation from Sparse Views." *arXiv preprint* arXiv:2408.10195 (2024).

**Questions:**

For Figure 9, did you use another multi-view (MV) model to convert the single-view input into MV images?

Line 309: Your proposed method also needs to 'maintain 3D consistency across input views.'

---

### Official Review · Reviewer_xURX · 2024-11-05

**Soundness:** 3
**Presentation:** 3
**Contribution:** 2
**Rating:** 5
**Confidence:** 5

**Summary:**

The paper proposes to reconstruct objects from randomly pose sparse views, rather than structured views. To deal with the problem, the authors propose to use Relative Coordinate Map, which is a pixel-level representation for cross-image correspondence. The RCM is used to aggregated predictions of each view in their own coordinate system into the world coordinate (the coordinate system of first view). The proposed method outperform previous LRMs that are designed for reconstructing structured views.

**Strengths:**

- The RCM extends the rigid 6D relative pose representation to a pixel-level representation, which improves the performance.
- The performance is better than prior works, e.g. LGM, iFusion, CRM, etc.
- The paper is well-written and is easy to follow.
- The method can be extended to 8 views.

**Weaknesses:**

- Missing related works. The RCM representation is an pxiel-level extension of relative pose between images. Thus, related works should be discussed [1,2,3,4] and the anthors should differentiate their work from previous works that use rigid pose for aggregation [2,3] for making the contribution more clear.

[1] Zhang, Jason Y., Deva Ramanan, and Shubham Tulsiani. "Relpose: Predicting probabilistic relative rotation for single objects in the wild." ECCV 2022.
[2] Jiang, Hanwen, et al. "Few-view object reconstruction with unknown categories and camera poses." 2024 International Conference on 3D Vision (3DV). IEEE, 2024.
[3] Lin, Amy, et al. "Relpose++: Recovering 6d poses from sparse-view observations." 3DV 2024.
[4] Zhang, Jason Y., et al. "Cameras as rays: Pose estimation via ray diffusion." ICLR 2024.

- Inappropriate baselines. I think the biggest problem of this paper is the inappropriate baselines. I agree that many of existing LRMs are designed for reconstruction from structured views at fixed camera pose, due to the use of the diffusion model which can only synthesize novel views at these fixed viewpoints. Thus, these models are biased to these fixed views. In contrast, the proposed model is actually trained on datasets with random viewpoints (as denoted in the paragraph of Line 766). This will naturally improve the performance on unstructured views. Thus, I don't believe the comparison with LGM, CRM, and InstantMesh is fair.

At the same time, I would like to point the authors to works for sparse-view reconstruction from unstructured views [2,5,6,7]. I can understand if the authors can not compare with PF-LRM [6] (which is also discussed in the paper) as their code is not released. But the authors can try to ask the authors of PF-LRM for providing their inference results for a potential comparison. The code of [2,5,6] are released and the authors should do a comparison, e.g. on the OmniObject3D dataset with [2,5] and on GSO with [6]. For this comparison, the authors should use the data and split of [2,5,6].

[5] Jiang, Hanwen, et al. "Leap: Liberate sparse-view 3d modeling from camera poses." ICLR 2024.
[6] Wang, Peng, et al. "Pf-lrm: Pose-free large reconstruction model for joint pose and shape prediction." ICLR 2024.
[7] Nagoor Kani, Bharath Raj, et al. "UpFusion: Novel View Diffusion from Unposed Sparse View Observations." ECCV 2024.

- More comparison. Although the author claim that SpaRP is a concurrent work, I hope the authors can compare with SpaRP for having a concept of the performance compared with a LRM, given the fact that the selected baselines are not appropriate. I can understand if the performance is not as good as SpaRP, but I encourage this performance comparison for future research.

- Missing evaluation. The RCM reveals the correspondence between images. Is it possible to compare with dense correspondence estimation methods for a comparison (on the shared observable parts)? I list some related wok here [8,9]. At least, the accuracy of RCM prediction should also be evaluated. The authors should report the L1/MSE distance between the prediction and the GT RCM, as well as reporting the correspondence accuracy (correspondence error in pixel). The authors should also include quantitative comparison with CCM.

[8] Edstedt, Johan, et al. "DKM: Dense kernelized feature matching for geometry estimation." CVPR 2023.
[9] Edstedt, Johan, et al. "RoMa: Robust dense feature matching." CVPR 2024.

- Suggested experiment. I encourage the authors to experiment with using ground-truth RCM. In this case, as the proposed method has a two stage pipeline, the experiment will enable us to understand its stability -- whether it is better than using rigid pose, and how robust it is to noisy estimation in the first stage. If the performance of using GT RCM is close to using predicted RCM, we can draw the conclusion that the RCM prediction is accurate and the method is robust enough to the noisy prediction.

In general, I believe this is an interesting paper. I can raise my score if the author discuss the related work, compare with more appropriate baselines and adding evaluation results of RCM.

**Questions:**

- Robustness to elevation. The authors seem to have a strict control of elevation. As illustrated in the paragraph of Line 766, the elevation is controlled with fixed angles of 20, -20, and 5 degree. Moreover, the real-world examples are all of images with small and almost fixed elevation. Based on this, I have two questions. First, is the model able to generalize to arbitrary elevation when they are trained on the fixed ones? This is important as real-world images have random elevations. Second, if we generate data with random elevation, will the performance be further improved (on both the current elevation setting and the random elevation setting)?

- Details of training and testing dataset. The authors should include the setting of elevation for rendering in the main text. This is important. Besides, the authors illustrate that they use 20-degree elevation images as inputs and evaluate on images with 5-degree images (Line 776). Why not just shuffle them randomly? I believe this evaluation setting is inappropriate. The authors should render images with random elevation between -20 to 20 degree for evaluation.

- Factual error. In Line 157, the authors claim "these methods need optimization". I am not sure whether PF-LRM is included, but actually PF-LRM doesn't require optimization.

---

### Note · Authors · 2024-11-14

I have read and agree with the venue's withdrawal policy on behalf of myself and my co-authors.